# Epidemiological Criminology and COVID: A Transdisciplinary Analysis of Violent Crime and Emergency Department Admissions during COVID

Lindsey Wylie [1], Julie Garman [2], Gaylene Armstrong [2], Ashley Farrens [3], Jennifer Burt [4], Mark Foxall [2], Michael Visenio [5], Macall Cox [5,6], Cynthia Hernandez [5,6], Charity Evans [5] and Ashley Ann Raposo-Hadley [5,*]

1  National Center for State Courts, Williamsburg, VA 23185, USA
2  School of Criminology and Criminal Justice, University of Nebraska at Omaha, Omaha, NE 68182, USA
3  Nebraska Medicine, Omaha, NE 68198, USA
4  Munroe-Meyer Institute, University of Nebraska Medical Center, Omaha, NE 68198, USA
5  Department of Surgery, University of Nebraska Medical Center, Omaha, NE 68198, USA
6  Department of Emergency Medicine, University of Nebraska Medical Center, Omaha, NE 68198, USA
*  Correspondence: ashley.raposohadley@unmc.edu; Tel.: +1-402-559-6111

**Abstract:** As little is known about the influence of COVID-19 on rates of violent crime, the purpose of this study is to examine violent injury captured by emergency department admissions and by law enforcement in a mid-sized midwestern city (Omaha, Nebraska) from January 2016 to December 2020. Although COVID-19 did not show a direct significant relationship, weeks during the COVID-19 period showed a marginal increase in incident rate ratios for violent incidents in both datasets. While violence remained stable during the pandemic, racial differences between samples were observed. This study emphasizes the utility of a transdisciplinary approach to understand the underlying drivers of violent crime and victimization.

**Keywords:** epidemiological criminology; COVID-19; pandemic; violence; emergency department; shootings

## 1. Introduction

Since October 2022, over 1,065,152 deaths have occurred in the United States as the direct result of COVID-19 [1]. Minimizing community spread of COVID-19 and enhancing public health during the pandemic has been a priority of many individuals and social institutions. Actions taken to reduce the spread of COVID-19 have impacted life across the globe with legal mandates, individual behavioral changes such as mask wearing, institutional policies such as social distancing, business and school closures, or the implementation of stay-at-home orders confining individuals and their families to their homes. While restrictions aimed to reduce viral transmission, some research suggests they have simultaneously contributed to unintended community consequences of violence [2].

Most research on crime after the designation of the COVID-19 pandemic has used official law enforcement data to assess changes in crime rates in geographically dense, populous urban cities [3,4]. Few studies have examined the impact of the pandemic on violence rates using a transdisciplinary lens in a Midwestern U.S. city. In this study, we combine criminological and medical data to explore the impact of the pandemic on violence in Omaha, Nebraska. Omaha is reflective of a significant proportion of U.S. cities given its population composition and characteristics, and research for small to medium cities in more rural states is currently limited [5]. In addition, as compared to research in other jurisdictions, Nebraska's governor did not issue a stay-at-home order in response to COVID-19. Instead, the state relied on public health officials who directed health measures that limited public gatherings.

As of 2020, in comparison to the national average, Nebraska's violent crime rates were consistently lower (300.9 violent crimes compared to the national average of 366.7 violence crimes in 2019). There was a slight downward trend in Omaha's homicide rate in the two years prior to the pandemic, although the city is not resistant to the burden of violence [6]. Although not central to the purpose of this study, as part of the data analysis, we acknowledge the civil rights protests occurring simultaneous to the pandemic. On 25 May 2020, George Floyd, a 46-year-old Black man in Minneapolis, Minnesota, was killed by a police officer while being arrested. Although the Black Lives Matter (BLM) movement is a global movement that began in 2013, the killing of George Floyd propelled months of protests and social justice advocacy calling for defunding of police and law enforcement reform. On 19 May 2020, BLM protests began in Omaha, Nebraska. These protests were further fueled by a local murder that resulted from a confrontation during a BLM protest on May 30th. During the evening BLM protest, a white bar owner who was armed and present at his business during the protest, shot and killed a 22-year-old activist who was Black after a confrontation and physical altercation. This murder resulted in further protesting and social disruption, as well as a city-wide curfew for several days.

### 1.1. COVID-19 Pandemic and Crime

After the onset of the pandemic, initial trends suggested crime rates decreased as much as 35% in major urban areas [3]. Yet, as the pandemic has continued, researchers collecting and analyzing additional data on crime rates have accounted for stay-at-home orders and other contextual factors. Findings from these more nuanced studies are mixed. While some studies found a decrease in crime rates, others found an increase, likely due to the operationalization of crime and varying geographical foci. For example, when crime is operationalized into violent or non-violent crimes in larger U.S. cities, researchers found pandemic restrictions contributed to a decline in nonviolent crime, but stability in violent crime [4,7,8]. When studies account for more specific crime types or geographical location, the effects of the pandemic and associated restrictions are less clear. For instance, Pietrawska, Aurand, and Palmer [9] compared crimes against persons with property crime in four cities over a period of 10 weeks and found substantial week-to-week variation in crime rates and types. Balkin and McDonald [10] reported that assault and battery decreased in New York, San Francisco, Los Angeles, Chicago, and Philadelphia during a two-week period in March 2020 when compared to the same time in 2019. Yet, when researchers singularly focused on Los Angeles, results demonstrated a decrease in battery crimes during March 2020, but no significant change in assault with a deadly weapon when compared to March 2017, 2018, or 2019 [11,12].

While the above studies inform the need to examine crime trends at the neighborhood-level in making conclusions about the pandemic effects, the existing research is also limited by the nature of the data utilized in analysis—namely, calls for service and official law enforcement data. These sources of crime data are vital to our understanding of crime rates, but scholars argue they may underreport crime during the pandemic. In addition, disruptions in daily activities for the population in general, such as working from home and online schooling, means that these data may not accurately capture victimization including intimate partner violence [2,13,14], which may not be reported to the police but could be detected in data on trauma admissions. In response, scholars have urged analysis that include other metrics to better understand trends in violence [5].

### 1.2. COVID-19 Pandemic and Hospital Trauma

With research demonstrating changes in crime rates during the COVID-19 pandemic, another metric for assessing change in violence is through examining hospital emergency department admissions. These admissions are different from standard Emergency Department (ED) visits and are categorized by patients sustaining injuries severe enough for potential surgical interventions and an increased length of stay in the hospital. In general, hospital admissions data suggest that in the early days of social restrictions, trauma activa-

tions (i.e., when trauma surgeons are called to arrive as quickly as possible) declined, driven in large part by fewer motor vehicle collisions and blunt trauma (i.e., non-penetrating injury) [15,16]; however, the proportion of penetrating trauma (e.g., gunshots and stabbings) and non-accidental trauma (i.e., intentional injury) appears to have increased in some trauma centers [15–18], suggesting that interpersonal violence has remained unchanged despite social restrictions.

For instance, Rhodes, Petersen, and Biswas [19] retrospectively reviewed all trauma registry patients to a rural Level I trauma center (a trauma center that provides the highest level of comprehensive care to traumatically injured patients) before and during the COVID-19 pandemic. They found that despite seeing a decrease in ED volume and overall trauma volume, there was a significant increase in domestic violence assaults that mostly presented as penetrating injuries. In a New Orleans, Louisiana Level I trauma center, analysis revealed fewer total trauma activations during the stay-at-home period; however, while the proportion of penetrating trauma was greater during the stay-at-home period, the proportion of blunt trauma was lower during the stay-at-home period than the predicted period. Furthermore, data indicated a shift from accidental blunt trauma to non-accidental penetrating trauma, and a greater increase in the proportion of gunshot wounds and assault injuries during the stay-at-home period than expected, despite no differences in the proportion of Tier 1 (i.e., critical medical interventions or surgery required shortly upon arrival to the hospital) and Tier 2 (i.e., injuries not immediately life threatening) activations [16].

In summary, aggregate information from hospital-based data sources indicated an overall reduction in violence in two cities. The challenge presented is that like earlier criminal justice studies, aggregate data may be masking more nuanced findings when considering specific neighborhoods and other contextual factors. The purpose of this study is to utilize a transdisciplinary approach to understanding violent crime rates in the city of Omaha, Nebraska. We consider changes in violent crime rates from the traditional criminological perspective using official law enforcement data and contrast the findings using hospital-based emergency department admission data for violent injury. This latter perspective stems from a traditional medicine or public health approach. Our analytical approach is informed by recent studies that underscore the importance of considering specific crime types, as well as disaggregated jurisdictions (i.e., less than city level) for analysis. As such, this study focused specifically on three types of violent crimes (i.e., criminal homicides, nonfatal shootings, and aggravated assaults) as these are more likely than other forms of violent crime (such as rape and sexual assault [20]) to be captured in official law enforcement data. In addition, previous research suggests stability in hospital admissions for these types of violent injury (e.g., gunshots, stabbings, and intentional injury) remained stable during the pandemic shutdown [16–18]. Further, utilizing both law enforcement and hospital data sources allows this study to capture incidents that are typically un-reported to police and expand data collection beyond the geographic area served by the hospital.

## 2. Materials and Methods

### 2.1. Data

Crime rate data from the Omaha Police Department and data for trauma hospital admissions from the Nebraska Medicine Electronic Health Records (EHR) database (EPIC) for the period spanning 2016 through 2020 were obtained. The Nebraska Medicine EHR only includes hospital admissions at Nebraska Medicine, the largest trauma center in the state and was entered into the system using International Classification of Disease codes (i.e., ICD10 codes). These data provided a five-year, three-month timeframe prior to the COVID-19 pandemic onset, and approximately nine-month period following the shutdown date of interest. Data from each source were cross-referenced using the victim's name (separated into three variables; first name, middle name, and last name) and the date of the incident (using a buffer of 24 h before and after to account for delayed reporting) to ensure duplicate cases were removed, and each incident was only counted once. Institutional

Review Board was obtained and a letter of support from the Omaha Police Department outlining the purpose for data use was provided.

### 2.1.1. Violent Injury Incidents

Within law enforcement data, there were 10,414 violent incidents documented. These incidents included criminal homicides ($n = 141$; 1.4%), nonfatal shootings ($n = 534$; 5.1%), and aggravated assaults ($n = 9739$; 93.5%). Criminal homicides included victims of intentional, knowing, reckless, or criminal negligence causing death. Nonfatal shootings included incidents where an individual was struck from a projectile of a gun in which the caliber was greater than a pellet gun. Aggravated assault cases were defined by the gravity of harm (compared to simple assault) and included incidents with a weapon or intent to commit a serious crime.

Within the hospital admission data, 3504 violent ED admissions were found; however, 67 cases were removed for not meeting inclusion criteria. Cases removed from the data included 11 cases (0.3%) who were follow-ups from an initial injury that presented to the ED and were removed once confirming the initial injury was captured in the data, and 56 cases (1.5%) that had non-violent injuries (i.e., suicide, complications from previous injuries, elder and child neglect). In addition, 176 duplicate hospital cases were identified and removed. The remaining 3261 hospital admission cases form the basis of our outcome measure of violent incidents. These violent incidents were categorized into three groups: assaults ($n = 2591$; 79.4%), stabbings ($n = 120$; 3.6%), and gunshot ($n = 550$; 16.8%) injuries. Assaults included assault with a blunt object, bodily force, glass, hot liquid, striking, pushing, unarmed brawling or fighting, human bite, asphyxiation, strangulation, and BB and pellet gun related mechanisms of injury. Stabbings included the use of sharp objects for stabbing or piercing, including razor blades, sharp glass, broken glass, and knives. Gunshots included wounds and assaults by gunshot.

Our primary variable of interest in the analysis is the pandemic onset or shutdown date. We include other available covariates that may have influenced violent injury incident rates, including the location of the incident and seasonality. Each of these variables and their operationalization are described in the following sections.

### 2.1.2. Pandemic Shutdown in Omaha, Nebraska

The independent variable used in this study is the COVID-19 pandemic shutdown, which we operationalized as incidents prior to the federal government declaring a national emergency on 13 March 2020, and incidents occurring on or after that date. The federal national emergency declaration marked the initial and substantial disruption to movement and behavior in Omaha (absent a state order in Nebraska or city order in Omaha). All incidents that occurred before 13 March 2020, were coded as pre-intervention incidents, and all incidents that occurred on or post 13 March 2020, were coded as post-intervention incidents. In Nebraska, specifically, the first state restrictions from the Department of Health and Human Services followed the federal declaration on 16 March 2020, with Omaha-area school closures beginning on 18 March 2020.

### 2.1.3. Civil Rights Protests

As discussed above, parallel to the pandemic, Omaha, Nebraska experienced months of civil rights activism disrupting typical social behavior patterns and approaches to policing. Like Kim and Phillips [21], a dummy variable reflecting the Black Lives Matter (BLM) movement the summer of 2020 was included to control for potential changes in violence as a result of disruption to policing. Ten observations occurred during the weeks of civil unrest between 25 May 2020, and 27 July 2020. These observations were coded as 1 and the rest were coded as 0.

### 2.1.4. Seasonality

Due to variation in climate in Nebraska and research that has found a positive association between seasons and crime [22,23], it is important to consider seasonality when analyzing violence data over time. To control for the effects of seasonality, a variable reflecting asymmetrical seasonal quarters was created, such that weeks within Q2 and Q3 were coded as 1, and weeks within Q1 and Q4 were coded as 0.

### 2.2. Analytical Model

Preliminary analyses used chi-square tests to measure differences in the proportion of each violent crime type, before and during the COVID-19 pandemic shutdown observational period. Alpha was set to 0.05. Pseudo R2 values were estimated using McFadden's Pseudo R2, sometimes called "deviance R2", calculated as one minus the ratio of the full-model log-likelihood to the intercept-only log-likelihood [24].

We used a negative binomial regression to examine the relation of the COVID-19 pandemic shutdown to weekly incidents of violence. Data were modeled from two separate sources: first, a dataset consisting of ED admissions; and second, a dataset consisting of police records for violent crime incidents. We assessed exponentiated model parameters displayed as incident rate ratios.

### 3. Results

### 3.1. Emergency Department Admissions

First, we examined the demographics of victims of violence admitted to the ED. The mean age of the hospital sample at the time of injury was 33.51 (SD = 14.59). The hospital sample was 59.9% male, with gunshots (84.2% male) and stabbings (80.8% male) driving the sex differences. The assault victims were relatively sex balanced (53.7% male). About half of the hospital sample (49.4%) was White/Caucasian, with Black/African American comprising the second largest racial ethnic group (36.7%). Although individuals who were White/Caucasian comprised the largest proportion of assault (52.1%) and stabbing (45.8%) victims, victims who were Black/African American comprised nearly half of all gunshot wounds (47.3%) (See Table 1).

**Table 1.** Victim Demographics and Violent Incidents for Emergency Department Hospital Admission Data.

| Variable | Assaults (*n* = 2591) | Gunshots (*n* = 550) | Stabbings (*n* = 120) | All (*n* = 3261) |
|---|---|---|---|---|
| Age, mean (SD) | 34.02 (14.17) | 30.85 (16.40) | 34.87 (13.25) | 33.51 (14.59) |
| Sex, N (%) male | 1392 (53.7) | 463 (84.2) | 97 (80.8) | 1952 (59.9) |
| Race, N (%) | | | | |
| White/Caucasian | 1349 (52.1) | 208 (37.8) | 55 (45.8) | 1612 (49.4) |
| Black/African American | 720 (27.8) | 260 (47.3) | 44 (36.7) | 1024 (31.4) |
| Other | 349 (13.5) | 55 (10.0) | 15 (12.5) | 419 (12.8) |
| American Indian or Alaskan Native | 114 (4.4) | 10 (1.8) | 3 (2.5) | 127 (3.9) |
| Asian | 19 (0.7) | 9 (1.6) | 1 (0.8) | 29 (0.9) |
| Unknown or patient refused | 17 (0.7) | 5 (0.9) | 2 (1.7) | 24 (0.7) |
| Hispanic | 13 (0.5) | 1 (0.2) | 0 | 14 (0.4) |
| Native Hawaiian or other Pacific Islander | 9 (0.3) | 2 (0.4) | 0 | 11 (0.3) |
| Ethnicity, N (%) | | | | |
| Not Hispanic or Latino | 2194 (84.7) | 487 (88.5) | 101 (84.2) | 2782 (85.3) |
| Hispanic or Latino | 381 (14.7) | 59 (10.7) | 18 (15.0) | 458 (14.0) |
| Unknown or patient refused | 16 (0.6) | 4 (0.7) | 1 (0.8) | 21 (0.6) |

### 3.2. Law Enforcement

Next, we examined the demographics of victims of violent crime reported through law enforcement data sources. The mean age of victims in the law enforcement sample was 32.87 (SD = 14.35). The law enforcement sample was 54.6% male. Criminal homicide and nonfatal shooting victims had the largest proportion of male victims, 77.3% and 79.2%, respectively, while aggravated assault victims were more sex balanced at 53.4% male. Contrary to the hospital sample, Black/African American victims were the largest racial and ethnic group in the police record derived sample, at 43.7% of all observations and comprising the largest racial and ethnic group of homicides (56.0%), nonfatal shootings (72.8%), and aggravated assaults (41.9%) (Table 2).

**Table 2.** Victim Demographics and Violent Incidents for Law Enforcement Data.

| Variable | Homicides (*n* = 141) | Nonfatal Shootings (*n* = 534) | Aggravated Assaults (*n* = 9739) | All (*n* = 10,386) |
|---|---|---|---|---|
| Age, mean (SD) | 32.97 (16.025) | 27.56 (12.210) | 33.16 (14.41) | 32.87 (14.35) |
| Sex, N (% male) | 109 (77.3) | 430 (79.2) | 5135 (53.4) | 5668 (54.6) |
| Race, N (%) | | | | |
| White/Caucasian | 36 (25.5) | 62 (11.6) | 3508 (36.0) | 3596 (34.6) |
| Black/African American | 79 (56.0) | 389 (72.8) | 4084 (41.9) | 4542 (43.7) |
| Hispanic | 15 (10.6) | 50 (9.4) | 999 (10.3) | 1059 (10.2) |
| Unknown | 7 (5.0) | 13 (2.4) | 580 (6.0) | 593 (5.7) |
| Native Hawaiian or Other Pacific Islander | 1 (0.7) | 6 (1.1) | 105 (1.1) | 111 (1.1) |
| American Indian or Alaskan Native | 3 (2.1) | 4 (0.7) | 207 (2.1) | 213 (2.1) |
| Asian | 0 | 2 (0.4) | 53 (0.5) | 55 (0.5) |
| Other | 0 | 9 (1.5) | 203 (2.1) | 211 (2.0) |

### 3.3. Type of Violence and the Pandemic Shutdown

3.3.1. Gun Violence

According to chi-square analysis, the proportion of gunshot wounds admitted to the ED did not significantly increase during the pandemic shutdown observational period as compared to before the shutdown (16.8% to 17.0%, respectively, $p = 0.927$). However, the proportion of gunshot injuries captured by law enforcement data sources significantly increased from prior to the shutdown to after the shutdown (from 6.0% to 7.3%, respectively, $p = 0.028$).

3.3.2. Stabbings

Chi-square analysis revealed that the proportion of stabbing admissions presented to the ED during the pandemic shutdown observational period was significantly higher than before the shutdown (3.2% to 6.1%, respectively, $p = 0.001$). Stabbings could not be analyzed separately using law enforcement data and are captured within the aggravated assault category.

3.3.3. Assaults

Chi-square analysis revealed that there was a non-significant decrease in the proportion of assault admissions to the ED during the pandemic shutdown observation period compared to before the shutdown (80.0% to 76.9%, respectively, $p = 0.097$). A similar significant trend was found in the law enforcement dataset, with the proportion of assaults decreasing from 94.0% to 92.7% ($p < 0.05$).

*3.4. Models Predicting Violent Incidents*

A negative binomial model was used to examine the relation of the COVID-19 pandemic shutdown on weekly ED admissions for violent injury. A binary variable reflecting seasonality and the BLM protest time-period were entered as covariates. Together, the predictors accounted for a small amount of the variance of the outcome, Pseudo $R2 = 0.002$, likelihood ratio $X2\ (3) = 5.181$, $p = 0.159$. The pandemic shutdown was not a significant predictor of weekly counts of violent ED admissions, while holding the BLM protest and quarterly seasonality constant. Although the pandemic shutdown time-period parameter was non-significant, there was a marginal increase in incidence rate ratio for weeks within the shutdown period compared to weeks before, IRR = 1.223, 95% CI [0.770–1.940] (Table 3).

**Table 3.** Negative Binomial Models, Incident Rate Ratios and 95% Confidence Intervals.

| Variable, Incident Rate (95% CI) | Emergency Department—All | Law Enforcement—All |
|---|---|---|
| Intercept | 11.451 (9.49, 13.82) *** | 31.958 (26.67, 38.29) *** |
| COVID-19 | 1.223 (0.77, 1.94) | 1.155 (0.80, 1.67) * |
| BLM | 1.356 (0.63, 2.92) | 1.247 (0.60, 2.57) |
| Quarter (2,3) | 1.133 (0.87, 1.48) | 1.356 (1.057, 1.74) * |

* $p < 0.05$, *** $p < 0.001$.

A negative binomial model was used to examine the same relationship with weekly incidents of violence captured by law enforcement. The BLM protest and seasonality variables were included as covariates again. Collectively, the predictors accounted for a significant amount of variance on the outcome, Pseudo $R2 = 0.003$, likelihood ratio $X2\ (3) = 8.992$, $p = 0.029$. The pandemic shutdown intervention period was not a significant predictor of the outcome while holding the BLM protest and quarterly seasonality constant. Similar to the hospital sample model, there was a slight increase in incident rate ratio for weeks within the shutdown period compared to weeks before, IRR = 1.155, 95% CI [0.798, 1.672].

**4. Discussion**

The results of this study should be viewed within the context of the study limitations. First, we were only able to include nine months of data for the time-period during the COVID-19 observation period from law enforcement and hospital trauma admissions sources. The limited data in the time following the shutdown may have weakened the ability to of models to isolate differences in violent victimizations. As time passes following the shutdown and as COVID-19 vaccines use becomes more widespread, communities have begun relaxing currently public health directives, which will provide additional opportunities to examine how violent crime may ebb and flow during such times. Second, data for this study was gathered from a mid-sized Midwestern city and therefore may not be generalizable to other geographical areas.

As the COVID-19 pandemic and associated health directives that limited public gatherings changed the way the world moved in the early months of 2020, this study employed a transdisciplinary approach to understand the effects of pandemic restrictions on violent crime rates in a mid-sized, Midwestern urban city. While Omaha, Nebraska was never under an official stay-at-home order, public health officials enacted directed health measures limiting public gatherings, which resulted in drastic restriction of most common everyday activities (e.g., school, business, and church closures, cancellation of public events, etc.) and reducing social movement within the city. Theoretically, restrictions such as these may have an effect on crime rates due to changes in the routine activities of the population, or as a result of variations in policing priorities and practices.

Existing research findings on the effect of the pandemic restrictions on crime rates in large urban areas are mixed. Scholars note that efforts to unpack the effects of stay-at-home orders on violent crime rates have been hampered by limitations in law enforcement data

sources such as a lack of neighborhood-level data and underreporting of certain violent crime types such as intimate partner violence. Findings from hospital ED admissions data on the effects of the shutdown on trauma activations were also mixed. While some findings suggested that trauma activations declined early during the shutdown (e.g., due to fewer motor vehicle collisions), others suggested both stability and increases in rates of other forms of trauma (e.g., penetrating trauma, non-accidental trauma, domestic violence assaults). Using both law enforcement and hospital emergency department data, we attempted to overcome these limitations to further understanding of the effects of the COVID-19 shutdown on specific violent crime types (e.g., gunshots, stabbings, assaults), while considering how seasonality and BLM protests may contextualize this relationship.

This study compared violent crime trends using five years of data on incidents reported to law enforcement and data on those admitted to a local hospital trauma center to examine trends related to the COVID-19 shutdown; specifically, the effect of public health restrictions on violent crime in Omaha, NE. Findings showed that in comparison to the period before the pandemic considered here, the proportion of gunshot injuries presenting to the ED remained virtually unchanged during the pandemic shutdown, although law enforcement data showed that the proportion of gunshot injuries increased in the time following the shutdown. Interestingly, stab injuries admitted to the ED were seen at a higher proportion during the COVID-19 time-period. At the health system level, the number of assaults presented for treatment remained unchanged, while law enforcement incidence of assault decreased during COVID-19 observation period, although these findings were not statistically significant. Accounting for seasonality and other events such as the BLM protests, social restrictions due to COVID-19 also do not appear to have significantly changed rates of violent crime in Omaha.

These findings align with other studies that examine the effect of pandemic restrictions on crime rates in the U.S. in that law enforcement and hospital ED admission data from Omaha suggest stability in violent crime trends in Omaha during the COVID-19 shutdown [4,7,13,16]. While cities have different demographics and sociopolitical factors driving crime rates, in a mid-sized Midwestern city in a predominantly rural state, COVID-19 was not a significant driving force in crime statistics as experienced elsewhere in the U.S. One potential explanation for the differences in these observations is the lack of an official stay-at-home order issued by local government. Although public health directives advised limited social gatherings and recommended avoiding prolonged contact with others, a mandated stay-at-home order would have further limited opportunities for interpersonal and community violence. Perhaps the health directive was sufficient to reduce opportunities for violent crime from what could have been a much higher rate of violence. Although not statistically significant, incident rate ratios for weeks during the COVID-19 observation period in all models were greater (above 1.0) compared to weeks before the observation period.

The data also reveal disparities between hospital and law enforcement demographics, such that a greater proportion of victims who presented to the hospital ED were White/Caucasian as compared to the law enforcement sample, where Black/African Americans comprised the largest racial and ethnic group. Trends in sex balance are similar between the two datasets. More males were victims of shootings, while total assault victims showed more sex balance in both hospital and law enforcement recorded shootings. Disparities between hospital and law enforcement demographic data may be related to individual decisions to seek treatment for violent injury such as financial concerns or personal health considerations related to seeking treatment during a pandemic.

The findings of this study help advance the limited research on the effects of the COVID-19 pandemic shutdown on changes in violent crime rates. First, the inclusion of hospital emergency department admissions data to supplement and contrast findings from existing official reports from law enforcement illustrates the utility of employing a transdisciplinary approach to understanding violent crime trends while taking greater account for the underreporting of domestic-related violence. Victims of violence who

were White/Caucasian were more likely to seek emergency medical services compared to victims who were Black/African American. Findings related to racial differences in trauma presentation were largely driven by domestic-related violent crimes, such as assault, meaning the inclusion of this data improves upon our understanding of who is victimized by violent crime in Omaha. Next, this study was able to account for how additional factors, such as seasonality and BLM protests interacted with violent crime rates both before and during the pandemic shutdown. Although our findings suggest these factors were not significant drivers of violence in Omaha, this may not be the case in larger urban or other areas.

Future research on the relationship between social restrictions and violent crime should examine how interruptions in routine activities may be more detrimental areas of a metropolitan area that are more at risk for violent perpetration and victimization.

**Author Contributions:** Conceptualization, A.A.R.-H., L.W., C.E., G.A. and J.B.; methodology, A.A.R.-H. and L.W.; formal analysis, A.A.R.-H. and L.W.; writing—original draft preparation, A.A.R.-H., L.W., J.G. and G.A.; writing—review and editing, C.E., A.A.R.-H., L.W., J.G., G.A., A.F., M.F., J.B., M.V., M.C. and C.H.; supervision—C.E. and G.A. All authors have read and agreed to the published version of the manuscript.

**Funding:** This research received no external funding.

**Institutional Review Board Statement:** The study was conducted in accordance with the Declaration of Helsinki and approved by the Institutional Review Board (or Ethics Committee) of University of Nebraska Medical Center (protocol code 0754-20-EP 1/29/2021).

**Informed Consent Statement:** Consents were not obtained or necessary since the hospital dataset was a retrospective electric health record review and the police data is publicly accessible.

**Data Availability Statement:** Not applicable.

**Acknowledgments:** The authors would like to thank the Omaha Police Department for their support of this research.

**Conflicts of Interest:** The authors declare no conflict of interest.

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
