# Peer review of "Epidemiological Criminology and COVID: A Transdisciplinary Analysis of Violent Crime and Emergency Department Admissions during COVID"

_traumacare, doi:10.3390/traumacare2040047_

Round 1
Reviewer 1 Report
This manuscript discusses hospital and law enforcement data of some forms of violent crimes for a short period of time at one location. As such, indicating that the findings "may not be generalisable" appears overstated. Other issues that should be addressed are mentioned below:
· The aim of the study should be clearly stated
· It should be stated whether or not ethical approval was obtained and how the researchers gained access to the data that were obtained (note that data is the plural form od ‘datum’ and requires plural rather than singular – grammatical error should be corrected in the manuscript)
· It should be made clear that there are numerous violent crime types and only a selection was used in the study – ensure a justification is provided
· In the discussion section it should be explained why a difference between hospital data and law enforcement data was obtained and whether this is supported by the existing literature
· There was no conclusion
Author Response
- This manuscript discusses hospital and law enforcement data of some forms of violent crimes for a short period of time at one location. As such, indicating that the findings "may not be generalisable" appears overstated.
We note only one reference to the findings not being generalizable to other geographical areas in the limitations on line 438.
- The aim of the study should be clearly stated
Thank you for this suggestion. We have more clearly articulated the purpose of the study in the abstract and introduction sections of the paper.
- It should be stated whether or not ethical approval was obtained and how the researchers gained access to the data that were obtained (note that data is the plural form od ‘datum’ and requires plural rather than singular – grammatical error should be corrected in the manuscript)
The manuscript has been checked to ensure all uses of the word “data” are plural. IRB review was obtained. A letter of support from the Omaha Police Department outlining the purpose of data use was obtained. This detail has been included in the manuscript for clarity.
- It should be made clear that there are numerous violent crime types and only a selection was used in the study – ensure a justification is provided
Thank you for this suggestion. Beginning on line 175, we have added context to articulate that these specific types of violent crime were examined as they are more likely to be captured in law enforcement data as well as trauma admissions. An additional citation was added to support this.
- In the discussion section it should be explained why a difference between hospital data and law enforcement data was obtained and whether this is supported by the existing literature
A discussion of the differences found between hospital data and law enforcement data is reported in the discussion section in lines 386-399.
Further, we have added context regarding reasons why demographic data between the two datasets varied. Disparities between hospital and law enforcement demographic data may be related to individual decisions to seek treatment for violent injury such as financial concerns or personal health considerations related to seeking treatment during a pandemic.
- There was no conclusion
The conclusion for this manuscript was included in the discussion section therefore a separate conclusion section was not added.
Reviewer 2 Report
This is a very strong article. It is clear and well-written and makes an important contribution to the literature. The combination of criminal justice data and public health data make this an especially significant contribution.
The literature review is very good and gives a good overview of the complexities of combining law enforcement information and information from emergency department admissions. The transdisciplinary analysis does indeed uncover and capture nuances, especially regarding negative events that do not come to the attention of the police. The ED data clearly show sex differences in violent victimization and importantly reveal the racial disparities in victimization. In criminal justice research it is of paramount importance to bring light to racial disparities, and the authors have done this explicitly, another strength of this work.
I do not have have statistical expertise and I hope another of the reviewers be helpful here.
The discussion is strong. While "stay at home" orders during the pandemic have been much-debated, this work reflects that they do have a strong impact on a community, perhaps both positive and negative. The article offers good suggestions for future research.
I have just a few suggestions:
The abstract should include that the study takes place in Omaha.
The first line of the article, line #28 and #29 on the number of Covid deaths needs to be updated. The authors can locate this information on the Centers for Disease Control and Prevention, Covid Data Tracker webpage.
In line # 143, "trauma surgery" should be replaced with "trauma surgeon"
Overall, an excellent effort!
Author Response
- The abstract should include that the study takes place in Omaha.
This has been updated to include Omaha, Nebraska, in the abstract.
- The first line of the article, line #28 and #29 on the number of Covid deaths needs to be updated. The authors can locate this information on the Centers for Disease Control and Prevention, Covid Data Tracker webpage.
Thank you for this suggestion. This figure has been updated in the manuscript.
- In line # 143, "trauma surgery" should be replaced with "trauma surgeon"
This has been updated to read “trauma surgeons.”